# MicroRNA analysis in maternal blood of pregnancies with preterm premature rupture of membranes reveals a distinct expression profile

**Michail Spiliopoulos**[1]*, **Andrew Haddad**[2], **Huda B. Al-Kouatly**[3], **Saeed Haleema**[4], **Michael J. Paidas**[1], **Sara N. Iqbal**[4], **Robert I. Glazer**[5]

1 Department of Obstetrics, Gynecology and Reproductive Sciences, Division of Maternal Fetal Medicine and Genetics, University of Miami, Miami, Florida, United States of America, 2 Department of Obstetrics, Gynecology & Women's Health, Division of Maternal Fetal Medicine & Surgery, Hackensack Meridian School of Medicine, Hackensack University Medical Center, Hackensack, New Jersey, United States of America, 3 Department of Obstetrics and Gynecology, Division of Maternal Fetal Medicine, Thomas Jefferson University, Philadelphia, Pennsylvania, United States of America, 4 Department of Obstetrics, Gynecology and Reproductive Sciences, Division of Maternal Fetal Medicine, MedStar Washington Hospital Center, Washington, DC, United States of America, 5 Department of Oncology and Lombardi Comprehensive Cancer Center, Georgetown University, Washington, DC, United States of America

* mxs3094@miami.edu

**Data Availability Statement:** All relevant data are within the paper and its Supporting information files.

## Abstract

### Objective

To determine the expression profile of microRNAs in the peripheral blood of pregnant women with preterm premature rupture of membranes (PPROM) compared to that of healthy pregnant women.

### Study design

This was a pilot study with case-control design in pregnant patients enrolled between January 2017 and June 2019. Patients with healthy pregnancies and those affected by PPROM between 20- and 33+6 weeks of gestation were matched by gestational age and selected for inclusion to the study. Patients were excluded for multiple gestation and presence of a major obstetrical complication such as preeclampsia, diabetes, fetal growth restriction and stillbirth. A total of ten (n = 10) controls and ten (n = 10) patients with PPROM were enrolled in the study. Specimens were obtained before administration of betamethasone or intravenous antibiotics. MicroRNA expression was analyzed for 800 microRNAs in each sample using the NanoString nCounter Expression Assay. Differential expression was calculated after normalization and log2- transformation using the false discovery rate (FDR) method at an alpha level of 5%.

### Results

Demographic characteristics were similar between the two groups. Of the 800 miRNAs analyzed, 116 were differentially expressed after normalization. However, only four reached FDR-adjusted statistical significance. Pregnancies affected by PPROM were characterized

**Funding:** The author(s) received no specific funding for this work.

**Competing interests:** The authors have declared that no competing interests exist.

by upregulation of miR-199a-5p, miR-130a-3p and miR-26a-5p and downregulation of miR-513b-5p (FDR adjusted p-values <0.05). The differentially expressed microRNAs participate in pathways associated with altered collagen and matrix metalloprotease expression in the extracellular matrix.

## Conclusion

Patients with PPROM have a distinct peripheral blood microRNA profile compared to healthy pregnancies as measured by the NanoString Expression Assay.

## Introduction

Premature rupture of membranes (PROM) is defined as rupture of the fetal membranes prior to the onset of labor and can occur at any gestational age, even as late as 42 weeks of gestation. Preterm PROM (PPROM) is defined as rupture of membranes prior to 37 weeks and complicates 2–4% of all singletons, and 7–20% of twin pregnancies. It is the most prevalent identifiable cause of preterm birth and responsible for 18–20% of perinatal deaths in the United States [1]. PPROM is a disease of the fetal membranes characterized by proteolysis, specifically directed to dismantle the structural integrity of the extracellular matrix (ECM) between the amnion and chorion, weakening the membranes.

Non-coding RNAs is a wide category of small and average size RNA molecules ranging from 22 nucleotides for microRNA to >200 nucleotides for long non-coding RNAs (lncRNA).

MicroRNAs (miRNAs) are highly conserved, single-stranded non-coding RNA molecules that regulate gene expression at the post-transcriptional level. More than 3,000 miRNA molecules are known thus far, and each of them can potentially target more than a hundred messenger RNA molecules. While a few miRNAs are specific to different tissues and organs, they are usually expressed in different combinations across diverse organs [2–6]. MiRNAs have a vast potential for regulation of cellular processes and are involved in cell differentiation, proliferation, apoptosis, metabolic homeostasis, as well as pathological processes such as infections [7], cardiovascular disease [8], and cancer [9–13]. Organ- or disease-specific miRNA expression patterns constitute miRNA profiles that may be used as diagnostic or prognostic clinical tools.

Studies have shown that miRNAs released into the circulation exist in a very stable form in contrast to messenger RNA molecules. Some miRNAs are released to the plasma within exosomes, which are circulating microparticles (50–100 nM in size) of endocytic origin which could facilitate intercellular communication. Exosomes from different cell types stimulate the immune system and have antitumorigenic effects. In contrast, exosomes released by tumor cells can suppress the immune system and enhance tumor progression [14–19].

Exosomes are also produced in the human placenta and have been shown to help in the establishment of maternal immune tolerance [20, 21]. Specific types of miRNAs are found in the plasma of pregnant women, with the plasma level of selected miRNA species altered depending on the gestational age. These observations support the role of miRNAs as extracellular messengers.

MiRNAs have also been investigated as possible biomarkers to identify women at risk for specific obstetrical outcomes including preeclampsia. Several miRNA subtypes have been identified to be significantly elevated in preeclampsia and gestational hypertension [22, 23]. Other studies have shown that specific profiles are present in ectopic pregnancy [24].

Our objective was to determine whether patients with PPROM have a unique serum miRNA profile compared to pregnancies without PPROM.

## Materials and methods

We performed a case-control pilot study between healthy pregnant patients (controls) and pregnant patients with PPROM (cases) between 20- and 33+6-weeks gestational age. Cases and controls were matched for gestational age. Our study was approved by the institutional review board of MedStar Health Research Institute (protocol# 2016–148). Patients with uncomplicated singleton pregnancies were recruited by trained physicians from either the low-risk obstetrics clinic or the midwifery clinic between January 2017 and June 2019. Patients with PPROM were recruited at the time of admission to the hospital. All patients in that group received betamethasone for fetal lung maturity and latency antibiotics according to hospital protocol. The diagnosis of PPROM was made by the attending physician at the time of admission and management during the hospital course was determined by the attending physician based on the PPROM protocol of our institution. After informed consent at the time of enrollment was obtained, specimen collection occurred at the time of initial blood draw after admission to the hospital for PPROM and before administration of betamethasone and latency antibiotics.

Patients were excluded for multiple gestation, stillbirth, fetal growth restriction (FGR), pre-eclampsia, major fetal anomalies, pregestational or gestational diabetes, chronic steroid use, acute or chronic immunologic disease, acute febrile illness at the time of presentation, cervical incompetence/cerclage in place, and inability or unwillingness of individual or legal guardian/representative to give written informed consent. A total of ten (n = 10) controls and ten (n = 10) patients with PPROM were enrolled in the study. Maternal blood was processed in order to obtain serum. All specimens were blinded to the lab performing the miRNA quantification.

### Statistical analysis

Statistical significance for baseline demographic variable associations with PPROM was determined with Chi-Square and Fisher's exact tests for categorical variables. Continuous variable associations with PPROM were assessed with the use of Mann-Whitney test. A p-value of $< 0.05$ was considered statistically significant.

### Specimen processing and total RNA isolation

Blood samples were drawn from our patients on admission to the hospital, serum was isolated by centrifugation (1600g, 15 minutes at 4˚C) from whole blood in less than 4 hours after collection and frozen immediately at -80 ˚C. Samples were checked for the presence of small clots and hemolysis. Specimens were maintained frozen for the duration of the study period and were shipped from Washington Hospital Center (Washington, DC, USA) directly to Georgetown University Department of Physiology (Washington, DC, USA) on dry ice where they remained blinded as to clinical outcome through testing, identified only by a unique identification number. MiRNA was isolated from serum samples using the miRNeasy Serum/Plasma Kit (Qiagen, Valencia, CA) according to the manufacturer's instructions.

### Nanostring analysis

MiRNA content and expression was analyzed for 800 human miRNAs on PPROM (n = 10) and control (n = 10) samples with the use of the NanoString nCounter miRNA/lncRNA

Expression Assay (NanoString Technologies Inc., Seattle, WA, USA) according to manufacturer's protocol. Sample preparation involves a multiplexed annealing of the specific tags to their target miRNA, a ligation reaction, and an enzymatic purification to remove the unligated tags. Sequence specificity between each miRNA and its appropriate tag is ensured by careful, stepwise control of annealing and ligation temperatures. Control RNA is included in the nCounter Human miRNA Sample Preparation Kit.

The nCounter miRNA expression assay is run on the nCounter Analysis system. The system is comprised of two instruments, the Prep Station used for post- hybridization processing, and the Digital Analyzer used for data collection. Data Collection is carried out in the nCounter Digital Analyzer. Digital images are processed, and the barcode counts are tabulated in a comma separated value (CSV) format. Statistical analysis was performed with the nSolver software, v4.0 (Nanostring Technologies, Seattle, WA). Raw counts were normalized using internal positive controls as well as housekeeping genes provided in the codeset as described in the nCounter gene expression data analysis guidelines [http://www.nanostring.com/applications/miRNA]. Background noise was assessed using negative control probes designed not to hybridize to any human target provided in the codeset. After normalization, log2-transformation and analysis, raw p-values for all miRNAs were corrected for multiple testing by the Benjamini–Hochberg false discovery rate [FDR] method at an alpha level of 5%. Only miRNAs with an FDR adjusted p-value of <0.05 were considered significantly altered.

## Bioinformatic miRNA target prediction

Predicted targets for the differentially expressed miRNAs were identified using the miRTargetLink version 2.0 software tool [25]. Gene Ontology [GO] and Kyoto Encyclopedia of Genes and Genomes [KEGG] pathway analyses were performed using the mirWalk 2.0 tool [http://www.ma.uniheidelberg.de/apps/zmf/mirwalk/] [26]. An interaction network for the differentially expressed miRNAs was created with use of strong validated targets from miRTarBase. We used miRPathDB version 2.0 to predict biological process pathways targeted by the differentially expressed miRNAs [27].

## Results

### Demographic characteristics

There were no statistically significant differences in the maternal age, race, BMI, parity, mode of delivery, umbilical cord pH and WBC count on admission between cases and controls (Table 1). The gestational age at blood draw was not significantly different between the two groups since patients were matched for this variable. There was a statistically significant difference in the gestational age at delivery for PPROM vs controls (29.3wks vs 39.4 wks, p<0.01), birth weight (1345 gms vs 3246 gms, p<0.01) and 1-minute Apgar scores (5.5 vs 8.3, p<0.01). In the PPROM group, 4 patients (40%) had a history of preterm delivery compared with no patients in the control group having such history. Three patients in the PPROM group (30%) went into spontaneous labor while two (20%) were induced. Indications for c-section in the PPROM group included placenta previa (10%), malpresentation (30%) and chorioamnionitis with non-reassuring fetal heart rate pattern (10%).

### Differential expression of miRNAs between PPROM and controls

Of the 800 miRNAs analyzed on the array, 116 showed differential expression between PPROM and control subjects with a statistical significance of <0.05 after normalization. However, only 4 miRNAs were significantly over- or under-expressed based on an FDR-adjusted

**Table 1. Demographic characteristics.**

|  | Controls (n = 10) | PPROM (n = 10) | p-value |
|---|---|---|---|
| Age (years) | 32.5±4.8 | 30.4±6.6 | 0.455 |
| BMI | 26.8±4.8 | 28.9±6.7 | 0.45 |
| Race |  |  | 0.851 |
| African-American (%) | 60 | 80 |  |
| Other (%) | 40 | 20 |  |
| GA at collection (wks) | 28.3±4.4 | 28.5±4.5 | 0.93 |
| GA at delivery (wks) | 39.4±1.3 | 29.2±4.5 | <0.01 |
| Nulliparous (%) | 40 | 30 | 0.32 |
| Mode of delivery |  |  | 0.175 |
| SVD (%) | 80 | 50 |  |
| Cesarean section (%) | 20 | 50 |  |
| Birth weight (gms) | 3246.2±561 | 1345.2±641 | <0.01 |
| Apgar 1 min | 8.3±1.1 | 5.5±2.9 | <0.01 |
| Apgar 5 min | 8.8±0.6 | 6.6±3.1 | 0.055 |
| Cord pH | 7.25±0.04 | 7.26±0.08 | 0.793 |
| WBC count (admission) | 9.28±2.9 | 9.8±2.7 | 0.705 |

Values given as average ± standard deviation

BMI = body mass index, GA = gestational age, SVD = spontaneous vaginal delivery, WBC = white blood cell count

p-value of <0.05 (Table 2). Three of the miRNAs (miR-513b-5p, miR-199a-5p and miR-130a-3p) had a >2-fold change in expression between PPROM and controls while one (miR-26a-5p) had a 1.86-fold change. miR-513b-5p was the only target to be significantly under-expressed in PPROM patients, while the other three were overexpressed. The remainder of miRNAs had an FDR-adjusted p-value of ≥0.05. Two additional miRNAs, miR-516b-5p and miR-526a, known to be part of the C19MC cluster on chromosome 19, expressed exclusively in the placenta, were under-expressed in the serum of PPROM patients (p-values of 0.004 and 0.006 respectively) but the FDR-adjusted p-value did not reach statistical significance (p = 0.33 and 0.43 respectively). The top 25 differentially expressed miRNAs based on FDR-adjusted p-value are shown on Table 3. Heatmaps of miRNA expression for all 800 targets and the top 25 differentially expressed miRNAs are shown in Figs 1 and 2 respectively.

**Pathways putatively targeted by the 4 microRNAs.** We used miRTargetlink 2.0 [25] to identify putative target pathways and genes for the four differentially expressed miRNAs that reached statistical significance. The top 20 significant biological process pathways for the network identified by miRPathDB 2.0 [27] are shown on Fig 3. Those with the highest statistical significance regulate the SMAD protein signal transduction pathway and the pathway-restricted SMAD protein phosphorylation. Regulation of T cell chemotaxis also appears to be a

**Table 2. MiRNAs with FDR adjusted p-value <0.05.**

| MicroRNA | Fold change | p-value | FDR adjusted p-value |
|---|---|---|---|
| miR-26a-5p | 1.86 | 0.00006 | 0.01 |
| miR-513b-5p | -2.4 | 0.0001 | 0.02 |
| miR-199a-5p | 2.31 | 0.0003 | 0.03 |
| miR-130a-3p | 2.64 | 0.0005 | 0.03 |

FDR = false discovery rate

**Table 3. Top 25 differentially expressed miRNAs.**

| MicroRNA | Fold change | p-value | FDR adjusted p-value |
|---|---|---|---|
| hsa-miR-26a-5p | 1.86 | 0.00006473 | 0.01 |
| hsa-miR-513b-5p | -2.4 | 0.00016399 | 0.02 |
| hsa-miR-199a-5p | 2.31 | 0.00034716 | 0.03 |
| hsa-miR-130a-3p | 2.64 | 0.00035399 | 0.03 |
| hsa-miR-30e-5p | -1.6 | 0.00052207 | 0.05 |
| hsa-miR-154-5p | -2.07 | 0.00070497 | 0.06 |
| hsa-let-7d-5p | 3.49 | 0.00085758 | 0.08 |
| hsa-miR-630 | -2.76 | 0.00123896 | 0.11 |
| hsa-miR-30c-5p | -2.44 | 0.00133779 | 0.11 |
| hsa-let-7g-5p | 1.94 | 0.00193919 | 0.16 |
| hsa-miR-126-3p | 3.02 | 0.0021172 | 0.18 |
| hsa-miR-129-5p | -3.03 | 0.00222641 | 0.18 |
| hsa-miR-199a-3p/hsa-miR-199b-3p | 2.77 | 0.00324963 | 0.26 |
| hsa-miR-125a-5p | 2.17 | 0.00330485 | 0.26 |
| hsa-let-7a-5p | 10.31 | 0.00342399 | 0.27 |
| hsa-miR-374b-5p | 1.65 | 0.003756 | 0.29 |
| hsa-miR-516b-5p | -1.85 | 0.00427739 | 0.33 |
| hsa-miR-151a-3p | 1.96 | 0.00455393 | 0.34 |
| hsa-miR-208b-5p | -1.96 | 0.00459024 | 0.34 |
| hsa-miR-107 | 1.75 | 0.00472435 | 0.35 |
| hsa-miR-491-5p | -2.4 | 0.00488638 | 0.36 |
| hsa-miR-191-5p | 2.48 | 0.00501527 | 0.36 |
| hsa-miR-323b-3p | -1.95 | 0.00593652 | 0.42 |
| hsa-miR-526a | -1.89 | 0.00616295 | 0.43 |

common pathway targeted by three of the index miRNAs. The interaction graph is shown in Fig 4. Molecular functions affected by the upregulated miRNAs were also identified and include protein kinase activity, nitric-oxide synthase activity and promoter sequence-specific DNA binding as shown in Fig 5.

Target pathways for miR-513b-5p do not appear to overlap with the remainder of the interaction network and include regulation of DNA-binding transcription factor activity as well as

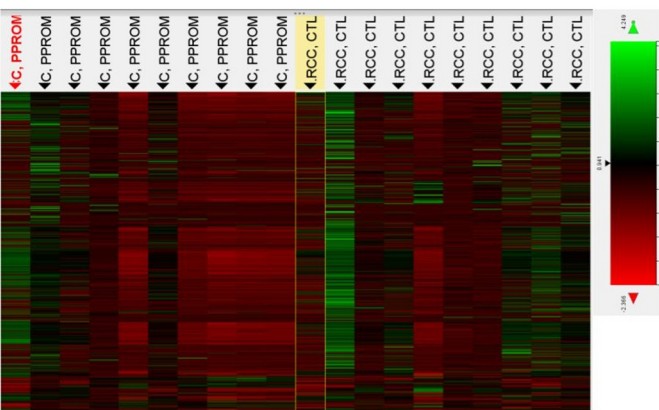

**Fig 1. Heatmap of miRNA expression in PPROM and control group for all 800 targets with color key.**

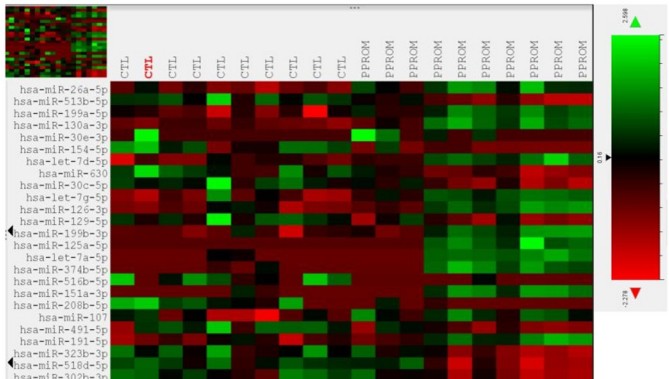

**Fig 2. Heatmap of miRNA expression for the top 25 differentially expressed miRNAs with color key.**

RNA biosynthesis. However, miR-513b-5p was recently shown to directly target Collagen type I alpha 1 chain (COL1A1) and COL1A2 to inhibit matrix metalloproteinase (MMP) pathways, thereby enhancing vascular smooth muscle cell death, apoptosis, inflammation and extracellular matrix destruction [28].

## Comment

**Principal findings.** The results of our study suggest that pregnancies affected by PPROM have a distinct serum miRNA profile that is characterized by upregulation of miR-199a-5p, miR-130a-3p and miR-26a-5p and downregulation of miR-513b-5p. In addition, placenta-derived miR-516b-5p and miR-526a were downregulated in PPROM vs controls but the fold-change did not reach FDR-adjusted statistical significance.

**Results in the context of what is known.** To our knowledge this is the first study to investigate miRNA expression in the serum of patients with PPROM. MiRNAs are known to target many genes through complex regulatory networks whose function remains to be elucidated. Downregulation of miR-513b-5p provided one of the strongest signals in our study. Interestingly, Zheng et al. [28] recently demonstrated that serum levels of miR-513b-5p were significantly decreased in patients with ruptured intracranial aneurysms compared to controls.

miEAA_results_ORA

| Category | Subcategory | Enrichment | P-value | P-adjusted | Q-value | Expected | Observed | miRNAs/precursors |
|---|---|---|---|---|---|---|---|---|
| GO Biological process (miRPathDB) | regulation of SMAD protein signal transduction | over-represented | 2.12541E-07 | 0.0001797034155 | 0.0001797 | 0.0210632 | 3 | hsa-miR-199a-5p; hsa-miR-26a-5p; hsa-miR-130a-3p |
| GO Biological process (miRPathDB) | regulation of pathway-restricted SMAD protein phosphorylation | over-represented | 1.21452E-07 | 0.0001797034155 | 0.0001797 | 0.0180542 | 3 | hsa-miR-199a-5p; hsa-miR-26a-5p; hsa-miR-130a-3p |
| GO Biological process (miRPathDB) | cardiac muscle cell development | over-represented | 1.73676E-06 | 0.00073421529 | 0.0007342 | 0.0391174 | 3 | hsa-miR-199a-5p; hsa-miR-26a-5p; hsa-miR-130a-3p |
| GO Biological process (miRPathDB) | positive regulation of pathway-restricted SMAD protein phosphorylation | over-represented | 1.33597E-06 | 0.00073421529 | 0.0007342 | 0.0361083 | 3 | hsa-miR-199a-5p; hsa-miR-26a-5p; hsa-miR-130a-3p |
| GO Biological process (miRPathDB) | ventricular septum morphogenesis | over-represented | 8.07654E-06 | 0.0027315 | 0.0027315 | 0.0631896 | 3 | hsa-miR-199a-5p; hsa-miR-26a-5p; hsa-miR-130a-3p |
| GO Biological process (miRPathDB) | T cell chemotaxis | over-represented | 1.81145E-05 | 0.0027931 | 0.0027931 | 0.00902708 | 2 | hsa-miR-199a-5p; hsa-miR-26a-5p |
| GO Biological process (miRPathDB) | branch elongation involved in mammary gland duct branching | over-represented | 3.62048E-05 | 0.0027931 | 0.0027931 | 0.0120361 | 2 | hsa-miR-26a-5p; hsa-miR-130a-3p |
| GO Biological process (miRPathDB) | cardiac muscle tissue growth | over-represented | 3.3132E-05 | 0.0027931 | 0.0027931 | 0.0992979 | 3 | hsa-miR-199a-5p; hsa-miR-26a-5p; hsa-miR-130a-3p |
| GO Biological process (miRPathDB) | cardiac septum morphogenesis | over-represented | 2.46547E-05 | 0.0027931 | 0.0027931 | 0.0902708 | 3 | hsa-miR-199a-5p; hsa-miR-26a-5p; hsa-miR-130a-3p |
| GO Biological process (miRPathDB) | endocardial cushion development | over-represented | 1.22909E-05 | 0.0027931 | 0.0027931 | 0.0722166 | 3 | hsa-miR-199a-5p; hsa-miR-26a-5p; hsa-miR-130a-3p |
| GO Biological process (miRPathDB) | long-term memory | over-represented | 3.62048E-05 | 0.0027931 | 0.0027931 | 0.0120361 | 2 | hsa-miR-199a-5p; hsa-miR-130a-3p |
| GO Biological process (miRPathDB) | peptidyl-lysine trimethylation | over-represented | 3.62048E-05 | 0.0027931 | 0.0027931 | 0.0120361 | 2 | hsa-miR-199a-5p; hsa-miR-26a-5p |
| GO Biological process (miRPathDB) | positive regulation of T cell chemotaxis | over-represented | 3.62048E-05 | 0.0027931 | 0.0027931 | 0.0120361 | 2 | hsa-miR-199a-5p; hsa-miR-26a-5p |
| GO Biological process (miRPathDB) | positive regulation of acute inflammatory response | over-represented | 3.62048E-05 | 0.0027931 | 0.0027931 | 0.0120361 | 2 | hsa-miR-199a-5p; hsa-miR-26a-5p |
| GO Biological process (miRPathDB) | positive regulation of chromosome organization | over-represented | 2.21892E-05 | 0.0027931 | 0.0027931 | 0.0872618 | 3 | hsa-miR-199a-5p; hsa-miR-26a-5p; hsa-miR-130a-3p |
| GO Biological process (miRPathDB) | positive regulation of epithelial to mesenchymal transition | over-represented | 3.012E-05 | 0.0027931 | 0.0027931 | 0.0962889 | 3 | hsa-miR-199a-5p; hsa-miR-26a-5p; hsa-miR-130a-3p |
| GO Biological process (miRPathDB) | positive regulation of histone modification | over-represented | 1.57887E-05 | 0.0027931 | 0.0027931 | 0.0782347 | 3 | hsa-miR-199a-5p; hsa-miR-26a-5p; hsa-miR-130a-3p |
| GO Biological process (miRPathDB) | positive regulation of ossification | over-represented | 2.72963E-05 | 0.0027931 | 0.0027931 | 0.0932798 | 3 | hsa-miR-199a-5p; hsa-miR-26a-5p; hsa-miR-130a-3p |
| GO Biological process (miRPathDB) | regulation of T cell chemotaxis | over-represented | 3.62048E-05 | 0.0027931 | 0.0027931 | 0.0120361 | 2 | hsa-miR-199a-5p; hsa-miR-26a-5p |
| GO Biological process (miRPathDB) | regulation of heart growth | over-represented | 3.63384E-05 | 0.0027931 | 0.0027931 | 0.102307 | 3 | hsa-miR-199a-5p; hsa-miR-26a-5p; hsa-miR-130a-3p |

**Fig 3. miRPathDB predicted biological process pathways for target miRNAs.**

COL1A1 and COL1A2 were shown to be direct targets of miR-513b-5p in patients with intracranial aneurysms. Downregulation of the index miRNA leads to overexpression of COL1A1 and COL1A2, enhanced MMP activity, abnormal collagen metabolism in the extracellular matrix (ECM) and possibly aneurysmal rupture. A similar increase in expression of collagen type I and increased MMP activity has been shown in amniotic membranes of patients with PPROM even in the absence of inflammation [29]. It is hypothesized that activation of MMPs in PPROM leads to increased collagen degradation in the amniotic membranes. Studies in rabbits also support that MMPs are involved in matrix remodeling as part of the healing process of PPROM [30]. Hence, overexpression of collagen type I might be an attempt to restore normal collagen levels in the ECM of fetal membranes. It is unclear whether decreased expression of miR-513b-5p could be causative to the mechanism leading to membrane rupture or reactive and part of the healing process.

The elevated levels of miR-119a-5p, miR-130a-3p and miR-26a-5p appear to have an effect on a large number of genes as shown in Fig 4. However, there is significant overlap of all three in the regulation of the SMAD signal transduction pathway. The SMAD signal transduction pathway and TGF-$\beta$ have been shown to regulate gene expression of type I collagen, COL1A1 and COL1A2 [31, 32].

MiR-516b-5p and miR-526a were downregulated in PPROM patients but did not reach FDR-adjusted statistical significance. They both belong to the C19MC cluster that is placenta specific. Hromadnikova et al. [33] also identified decreased expression of the above miRNAs in the placentas of patients with PPROM. The lack of statistical significance in our study could

**Fig 4. miRTargetLink 2.0—Interactive miRNA target gene and target pathway networks.**

| Gene Ontology (miRWalk) | GO0014033 neural crest cell differentiation | over-represented | 0.000134026 | 0.0451068 | 0.0451068 | 0.158433 | 3 | hsa-miR-199a-5p; hsa-miR-26a-5p; hsa-miR-130a-3p |
|---|---|---|---|---|---|---|---|---|
| Gene Ontology (miRWalk) | GO0032525 somite rostral caudal axis specificati | over-represented | 4.59177E-05 | 0.0280098 | 0.0280098 | 0.112436 | 3 | hsa-miR-199a-5p; hsa-miR-26a-5p; hsa-miR-130a-3p |
| Gene Ontology (miRWalk) | GO0032909 regulation of transforming growth fac | over-represented | 0.000178423 | 0.0466449 | 0.0466449 | 0.173765 | 3 | hsa-miR-199a-5p; hsa-miR-26a-5p; hsa-miR-130a-3p |
| Gene Ontology (miRWalk) | GO0048733 sebaceous gland development | over-represented | 4.59177E-05 | 0.0280098 | 0.0280098 | 0.112436 | 3 | hsa-miR-199a-5p; hsa-miR-26a-5p; hsa-miR-130a-3p |
| Gene Ontology (miRWalk) | GO0048859 formation of anatomical boundary | over-represented | 3.96562E-05 | 0.0280098 | 0.0280098 | 0.107325 | 3 | hsa-miR-199a-5p; hsa-miR-26a-5p; hsa-miR-130a-3p |
| Gene Ontology (miRWalk) | GO0051098 regulation of binding | over-represented | 0.000147891 | 0.0451068 | 0.0451068 | 0.163543 | 3 | hsa-miR-199a-5p; hsa-miR-26a-5p; hsa-miR-130a-3p |
| Gene Ontology (miRWalk) | GO0071559 response to transforming growth fac | over-represented | 0.00010895 | 0.0451068 | 0.0451068 | 0.148211 | 3 | hsa-miR-199a-5p; hsa-miR-26a-5p; hsa-miR-130a-3p |

**Fig 5. miRWalk GeneOntology database prediction for target miRNAs.**

be the result of sampling maternal serum compared to reproductive tissues such as placenta or cervix or due to the relatively small sample size.

Several other studies in the literature have reported unique miRNA profiles in the placenta of patients with PPROM, preterm birth and other pregnancy complications such as pre-eclampsia and fetal growth restriction [33–35]. Elovitz et al. [36] examined miRNA expression in cervical cells of patients destined to have preterm birth and identified a distinct profile in this group. However, maternal serum miRNA signatures in the same group did not yield statistically significant results leading to the conclusion that local miRNA panels targeting reproductive tissues might prove more beneficial for discovering biomarkers or understanding of disease pathophysiology.

**Clinical implications.** Our findings shed some light in the pathophysiology underlying PPROM. ECM remodeling, altered collagen metabolism and increased MMP in fetal membranes have been reported in PPROM [29, 30]. Altered miRNA expression as shown in our study could be one of the mediators for these effects.

Specimen collection in our study was performed after the diagnosis of PPROM, hence it is unclear whether the distinct miRNA signature identified can be used as a potential biomarker for prediction of PPROM in otherwise asymptomatic patients. Nevertheless, the altered expression of miRNAs and the downstream effects on their putative gene targets are consistent with known pathways that have been previously implicated in PPROM. Since miRNAs from reproductive tissues, such as amniotic membranes and the placenta, can be released into the maternal circulation, it would be anticipated that local differences in expression can be detected systemically, but likely at lower expression levels than in placenta or cervical tissue. This release could be enhanced with interruption of the maternal-fetal interface such as in PPROM.

**Research implications.** Considering that our tissue sample was maternal serum and not amniotic membranes it would be important for future research to validate the above miRNA findings in reproductive tissues. In addition, future research on amniotic membranes could confirm or refute whether cell transfection with the above miRNAs alters mRNA expression for COL1A1 and COL1A2 by up- or down-regulation of their respective pathways.

**Strengths and limitations.** Our study strengths include: 1) all specimens were blinded to the lab performing the quantification of miRNA levels, 2) patients in the control and PPROM groups were matched by gestational age, 3) we used an established microarray technology for quantification of miRNAs, 4) specimen collection for the PPROM group was performed before the administration of betamethasone and latency antibiotics, thus ruling out any possible effect on the miRNA expression profile.

Our study was limited by recruiting patients after PPROM was diagnosed, hence not allowing for conclusions regarding the prognostic ability of the miRNA profile in PPROM. However, our primary goal was to elucidate pathways associated with PPROM pathophysiology rather than the identification of predictive biomarkers. In addition, our study lacked a validation cohort for the differentially expressed miRNAs. Even though the use of the false discovery rate for statistical significance decreased the likelihood of false-positive and false-negative

results, other modalities such as qRT-PCR might identify different miRNA expression levels. We also did not sample miRNAs in reproductive tissues such as amniotic membranes or cervix, that some researchers have proposed as preferable specimens to peripheral blood. Finally, the number of specimens obtained was limited.

## Conclusion

In summary, our findings suggest that patients with PPROM have a distinct serum miRNA profile compared to controls. Exploring local miRNA expression in reproductive tissues from patients before and after PPROM is diagnosed will be able to shed more light into its pathophysiology and possibly identify prognostic and therapeutic targets.

## Supporting information

**S1 Data.**
(XLSX)

## Author Contributions

**Conceptualization:** Michail Spiliopoulos, Sara N. Iqbal.

**Data curation:** Michail Spiliopoulos.

**Formal analysis:** Michail Spiliopoulos, Robert I. Glazer.

**Funding acquisition:** Michail Spiliopoulos.

**Investigation:** Michail Spiliopoulos, Robert I. Glazer.

**Methodology:** Michail Spiliopoulos.

**Project administration:** Michail Spiliopoulos, Andrew Haddad, Sara N. Iqbal, Robert I. Glazer.

**Resources:** Michail Spiliopoulos, Andrew Haddad, Saeed Haleema, Robert I. Glazer.

**Software:** Michail Spiliopoulos.

**Supervision:** Michail Spiliopoulos.

**Validation:** Michail Spiliopoulos.

**Visualization:** Michail Spiliopoulos.

**Writing – original draft:** Michail Spiliopoulos.

**Writing – review & editing:** Michail Spiliopoulos, Andrew Haddad, Huda B. Al-Kouatly, Michael J. Paidas, Sara N. Iqbal, Robert I. Glazer.

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
