## [Decision Letter · Decision Letter 0]

25 Jul 2022

PONE-D-22-15156MicroRNA analysis in maternal blood of pregnancies with preterm premature rupture of membranes reveals a distinct expression profilePLOS ONE

Dear Dr. Spiliopoulos,

Thank you for submitting your manuscript to PLOS ONE. After careful consideration, we believe that it has merit but does not fully meet PLOS ONE’s publication criteria as it currently stands. Therefore, we invite you to submit a revised version of the manuscript that addresses the points raised during the review process.

Please address the Reviewer's concerns point-by-point in your response. In addition, please include the requested clinical information in the revised version, acknowledge the limitations (e.g., cause or effect of PPROM, need for a validation cohort) and adjust the conclusions accordingly.

We look forward to receiving your revised manuscript.

Kind regards,

Tamas Zakar

Academic Editor

PLOS ONE

Journal Requirements:

Reviewers' comments:

Reviewer's Responses to Questions

**Comments to the Author**

1. Is the manuscript technically sound, and do the data support the conclusions?

Reviewer #1: Yes

Reviewer #2: Yes

2. Has the statistical analysis been performed appropriately and rigorously? 

Reviewer #1: Yes

Reviewer #2: Yes

3. Have the authors made all data underlying the findings in their manuscript fully available?

Reviewer #1: Yes

Reviewer #2: Yes

4. Is the manuscript presented in an intelligible fashion and written in standard English?

Reviewer #1: Yes

Reviewer #2: Yes

5. Review Comments to the Author

Reviewer #1: The results from this study have shown that there is a differential expression of specific miRNAs between PPROM and healthy controls in a small number of samples. The number of samples is very limited and due to the PPROM samples being collected at the time of hospital admission, it is not possible to determine whether these miRNAs are predictive or diagnostic of PPROM.

As it is not possible to distinguish the differential expression of miRNAs that could potentially lead to PPROM and those that are caused by PPROM, the bioinformatics work identifying potential targets of these miRNA markers would not be meaningful.

The study needs to include a validation cohort to further examine the differential expression of the identified miRNAs.

In addition, there is lack of information about how the samples were collected and processed. This is important to include as various factors such as phlebotomy variables, storage conditions/times, blood cell content, presence of clots/haemolysis, and processing can affect miRNA expression in serum.

There is important information on patient characteristics that is missing. Most of the patients were multiparous, but no information is given whether they previously had a term or preterm deliveries. It is also helpful to describe whether the cohort of patients went into spontaneous labour with PPROM or whether they were induced and some further information to explain why there is a 50% caesarean section rate for these patients.

Reviewer #2: This study is well-written and tried to find answer for PPROM pathogenesis. The material method is well defined and the results are presented accurately.It has originality and have interest inn this field. and It can be published.

6. PLOS authors have the option to publish the peer review history of their article (what does this mean?). If published, this will include your full peer review and any attached files.

Reviewer #1: No

Reviewer #2: No

---

## [Author Response · Author response to Decision Letter 0]

22 Aug 2022

Response to Reviewers:

Reviewer #1:

“The results from this study have shown that there is a differential expression of specific miRNAs between PPROM and healthy controls in a small number of samples. The number of samples is very limited and due to the PPROM samples being collected at the time of hospital admission, it is not possible to determine whether these miRNAs are predictive or diagnostic of PPROM. 

As it is not possible to distinguish the differential expression of miRNAs that could potentially lead to PPROM and those that are caused by PPROM, the bioinformatics work identifying potential targets of these miRNA markers would not be meaningful.”

We agree with the above reviewer comment regarding our study design not being able to differentiate between predictive and diagnostic markers for PPROM. This limitation is stated in both the “Clinical implications” as well as “Limitations” sections of the manuscript. 

However, as noted by the title and conclusion of our study, our primary goal was to shed some light in the pathophysiology of PPROM rather than attempt to identify predictive biomarkers. Hence, we believe there is value in identifying targets for the differentially expressed miRNAs.

Manuscript text and conclusions have been modified accordingly.

“The study needs to include a validation cohort to further examine the differential expression of the identified miRNAs.”

We acknowledge the lack of a validation cohort in our study. Even though the criteria set for FDR in our microarray study were very stringent there is still a possibility that qRT-PCR might identify different expression levels for the differentially expressed miRNAs.

Manuscript was edited to acknowledge the above.

“There is important information on patient characteristics that is missing. Most of the patients were multiparous, but no information is given whether they previously had a term or preterm deliveries. It is also helpful to describe whether the cohort of patients went into spontaneous labour with PPROM or whether they were induced and some further information to explain why there is a 50% caesarean section rate for these patients.”

In the PPROM group, 4 patients (40%) had a history of preterm delivery compared with no patients in the control group having such history. Three patients in the PPROM group (30%) went into spontaneous labor while two (20%) were induced. Indications for c-section (50%) in the PPROM group included placenta previa (10%), malpresentation (30%) and chorioamnionitis with non-reassuring fetal heart rate pattern (10%). 

The above information was added to the manuscript. 

“In addition, there is lack of information about how the samples were collected and processed. This is important to include as various factors such as phlebotomy variables, storage conditions/times, blood cell content, presence of clots/haemolysis, and processing can affect miRNA expression in serum.”

Blood samples were drawn from our patients on admission to the hospital, serum was isolated by centrifugation (1600g, 15 minutes at 4°C) from whole blood in less than 4 hours after collection and frozen immediately at -80 °C. Samples were checked for the presence of small clots and hemolysis. Specimens were maintained frozen for the duration of the study period and were shipped from Washington Hospital Center (Washington, DC, USA) directly to Georgetown University Department of Physiology (Washington, DC, USA) on dry ice where they remained blinded as to clinical outcome through testing, identified only by a unique identification number.

We included the above in the manuscript.

“Please address the Reviewer's concerns point-by-point in your response. In addition, please include the requested clinical information in the revised version, acknowledge the limitations (e.g., cause or effect of PPROM, need for a validation cohort) and adjust the conclusions accordingly.”

Above comments from the Academic Editor were addressed in the manuscript as well.

---

## [Decision Letter · Decision Letter 1]

19 Oct 2022

MicroRNA analysis in maternal blood of pregnancies with preterm premature rupture of membranes reveals a distinct expression profile

PONE-D-22-15156R1

Dear Dr. Spiliopoulos,

We’re pleased to inform you that your manuscript has been judged scientifically suitable for publication and will be formally accepted for publication once it meets all outstanding technical requirements.

Kind regards,

Tamas Zakar

Academic Editor

PLOS ONE

Additional Editor Comments (optional):

Reviewers' comments:

Reviewer's Responses to Questions

**Comments to the Author**

1. If the authors have adequately addressed your comments raised in a previous round of review and you feel that this manuscript is now acceptable for publication, you may indicate that here to bypass the “Comments to the Author” section, enter your conflict of interest statement in the “Confidential to Editor” section, and submit your "Accept" recommendation.

Reviewer #2: All comments have been addressed

2. Is the manuscript technically sound, and do the data support the conclusions?

Reviewer #2: Yes

3. Has the statistical analysis been performed appropriately and rigorously? 

Reviewer #2: Yes

4. Have the authors made all data underlying the findings in their manuscript fully available?

Reviewer #2: Yes

5. Is the manuscript presented in an intelligible fashion and written in standard English?

Reviewer #2: Yes

6. Review Comments to the Author

Reviewer #2: (No Response)

7. PLOS authors have the option to publish the peer review history of their article (what does this mean?). If published, this will include your full peer review and any attached files.

Reviewer #2: No

---

## [Editor Report · Acceptance letter]

24 Oct 2022

PONE-D-22-15156R1 

MicroRNA analysis in maternal blood of pregnancies with preterm premature rupture of membranes reveals a distinct expression profile 

Dear Dr. Spiliopoulos:

I'm pleased to inform you that your manuscript has been deemed suitable for publication in PLOS ONE. Congratulations! Your manuscript is now with our production department. 

Kind regards, 

on behalf of

Dr Tamas Zakar 

Academic Editor

PLOS ONE